# A Fast and Accurate Obstacle Segmentation Network for Guava-Harvesting Robot via Exploiting Multi-Level Features

**Jiayan Yao [1], Qianwei Yu [1], Guangkun Deng [1], Tianjun Wu [1], Delin Zheng [1], Guichao Lin [1,2,*], Lixue Zhu [1,2,*] and Peichen Huang [3]**

1   School of Mechanical and Electrical Engineering, Zhongkai University of Agriculture and Engineering, Guangzhou 510225, China
2   Guangdong Laboratory for Lingnan Modern Agriculture, Guangzhou 510642, China
3   College of Automation, Zhongkai University of Agriculture and Engineering, Guangzhou 510225, China
*   Correspondence: guichaolin@126.com (G.L.); zhulixue@zhku.edu.cn (L.Z.)

**Abstract:** Guava fruit is readily concealed by branches, making it difficult for picking robots to rapidly grip. For the robots to plan collision-free paths, it is crucial to segment branches and fruits. This study investigates a fast and accurate obstacle segmentation network for guava-harvesting robots. At first, to extract feature maps of different levels quickly, Mobilenetv2 is used as a backbone. Afterwards, a feature enhancement module is proposed to fuse multi-level features and recalibrate their channels. On the basis of this, a decoder module is developed, which strengthens the connection between each position in the feature maps using a self-attention network, and outputs a dense segmentation map. Experimental results show that in terms of the mean intersection over union, mean pixel accuracy, and frequency weighted intersection over union, the developed network is 1.83%, 1.60% and 0.43% higher than Mobilenetv2-deeplabv3+, and 3.77%, 2.43% and 1.70% higher than Mobilenetv2-PSPnet; our network achieved an inference speed of 45 frames per second and 35.7 billion floating-point operations per second. To sum up, this network can realize fast and accurate semantic segmentation of obstacles, and provide strong technical and theoretical support for picking robots to avoid obstacles.

**Keywords:** picking robot; semantic segmentation; obstacle segmentation; Mobilenetv2

## 1. Introduction

Guava is a famous fruit of southern China, and there is a big growing area and a huge labor force needed to harvest the fruit. The price of manual picking has increased due to the factors of China's aging population and the decline in rural youth. Therefore, the research on the guava-picking robots [1] to replace manual work has attracted people's attention. Fruit trees grow randomly and widely in an unstructured setting, making it easy for the branches to hurt the robot arms when picking. Based on the foregoing elements, it is crucial to segment the obstacles, such as fruits and branches.

Numerous research on fruit and branch recognition has been conducted recently [2]. Traditional techniques detect them by analyzing their texture and color characteristics. For example, Gongal et al. [3] used textural feature to detect apples; and Amatya et al. [4,5] classified cherry canopy pixels using a color-based Bayes classifier and achieved an 89% segmentation accuracy. However, these techniques are only effective when the pixel features are apparent and the picking environment is simple. In recent years, many excellent semantic segmentation models have been proposed, such as Encnet [6], SKnet [7], Upernet [8], CCnet [9], and so on. These models are frequently applied in the segmentation of orchard obstacles. For example, Zhang et al. [10] used Deeplabv3 [11] to segment the branches and trunks of apple trees. Yang et al. [12] used Mask R-CNN to detect fruits and branches of citrus trees. Chen et al. [13] used Deeplabv3+, U-net [14] and pix2pix [15], respectively, to segment the blocked branches. Chen et al. [16] proposed a lightweight network Sandglass-MFN based on hourglass structure and multi-feature fusion to segment

banana stem. Peng et al. [17] used Deeplabv3+ with Xception65 as the backbone to realize semantic segmentation of litchi branches. Majed et al. [18] used Segnet to segment the trunks and branches of apple trees.

Although being widely employed in the field of agricultural robotics to segment obstacles, these semantic segmentation networks still have certain issues. For example, PSPnet [19] missed a lot of detailed information in its encoder; Deeplabv3+ used a simple up-sampling operation to output a dense segmentation map, which caused the prediction result to be too coarse; ERFnet [20] made a compromise between the module's running speed and precision, though some of the model's accuracy was still lost. Based on the above problems, this research focuses on proposing a fast and accurate obstacle segmentation network. Specifically, Mobilenetv2 [21] is used as the backbone to output different levels of features. Afterwards, a feature enhancement module (FEM) is proposed to fuse and recalibrate these features. At last, a decoder based on self-attention is used to strengthen the connections between the positions in the features, and produce a compact segmentation feature map.

The main contributions of this paper are as follows:

1. In order to exploit multi-level features, a feature enhancement network, termed FEM, is proposed. FEM fuses different levels of feature maps, and uses a simple attention mechanism to recalibrate the channels of the feature maps, thus outputting a feature map with strong semantics and details.
2. In order to improve the segmentation accuracy, a novel decoder is proposed. It uses a self-attention layer to capture long-range dependency for each pixel, and utilizes a shortcut connection and element-wise addition to promote the gradient to flow in the network.
3. The method has a good segmentation performance. The mean intersection over union (MIOU), mean pixel accuracy (MPA) and frequency weighted intersection over union (FWIOU) are 76.30%, 84.63% and 89.04% respectively. The number of parameters, floating-point operations per second (FLOPs) and frames per second (FPS) are 3.9 million, 35.7 billion and 45, respectively.

## 2. Materials and Methods

### 2.1. Image Acquisition and Annotation

The images were collected on 24 September 2021 at Jiashuo Farm in Gull Island, Guangzhou City, Guangdong Province, China. The weather was fine that day, and the collection time was from 12 o'clock to 16 o'clock. An Intel RealSense D435i depth camera was used to take guava pictures with a resolution of $480 \times 640$ pixels at 30 frames per second. The distance between the guava tree and the camera was about 0.6 m. The row spacing of guava trees in the farm was 3.1 m, and the plant spacing was 2.5 m. In order to capture images from various perspectives, people with cameras went back and forth in the middle of the aisle, and up and down along the tree heights during the capturing process. In the end, about 40,000 images were obtained. Some images are shown in Figure 1.

A total of 891 images were randomly selected as the guava dataset. The train set, validation set, and test set ratios for this guava dataset were 6:1:3. Each image in the dataset was labeled pixel-by-pixel using an open-source software LabelMe 3.16.7. Every pixel in the image was labeled as a fruit, branch or background object. An example is shown in Figure 2.

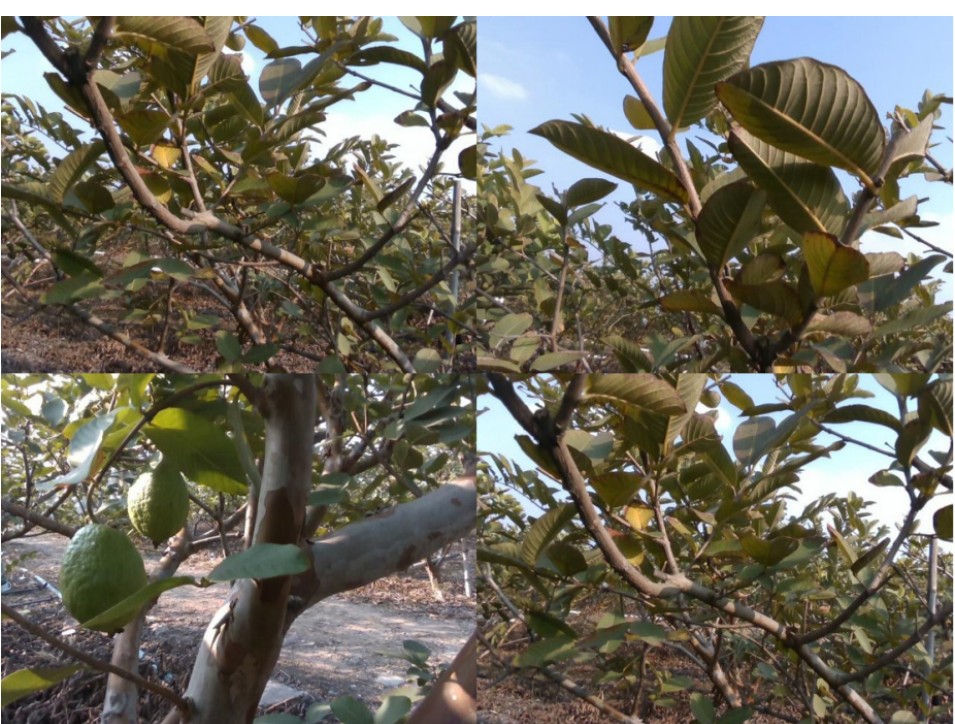

**Figure 1.** Examples illustrating the captured images.

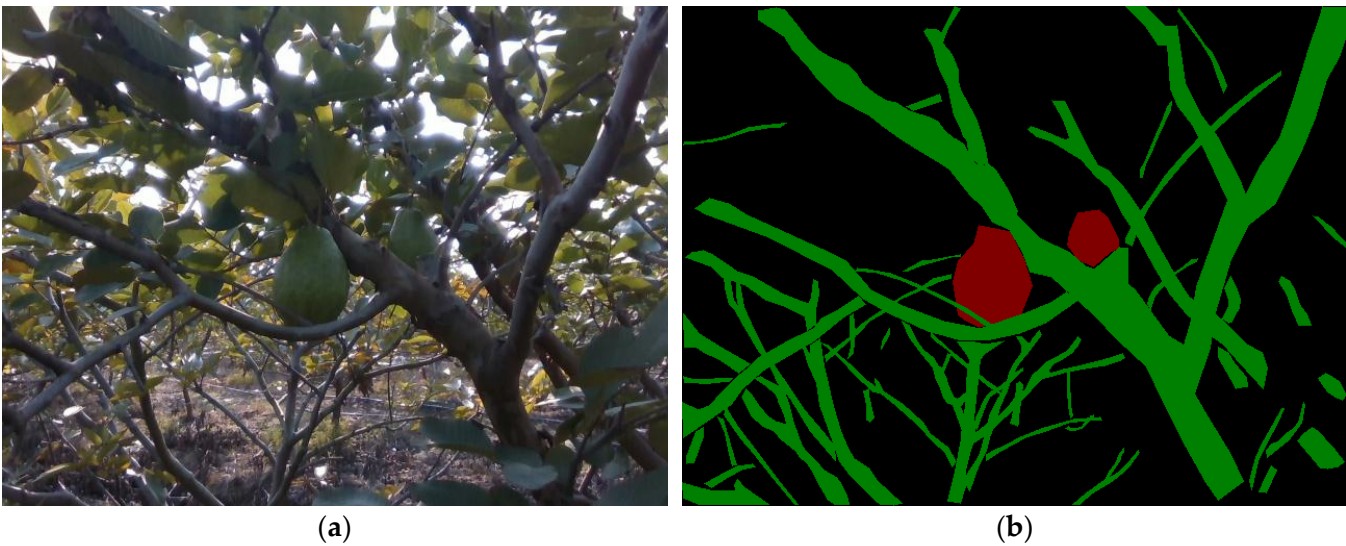

| (**a**) | (**b**) |

**Figure 2.** A captured image (**a**) and its corresponding annotation (**b**).

### 2.2. Data Augmentation

Data augmentation is an important process that is useful in many different domains, including object identification, semantic segmentation, image classification and so on. In general, proper data augmentation can strengthen robustness of the model to attain higher accuracy. The common data augmentation techniques include panning, flipping, adjusting image contrast and saturation, and so on. Random cropping and left-right mirroring are employed in this work. More augmentation methods will be used in the future to broaden the datasets. Figure 3 shows a visual example.

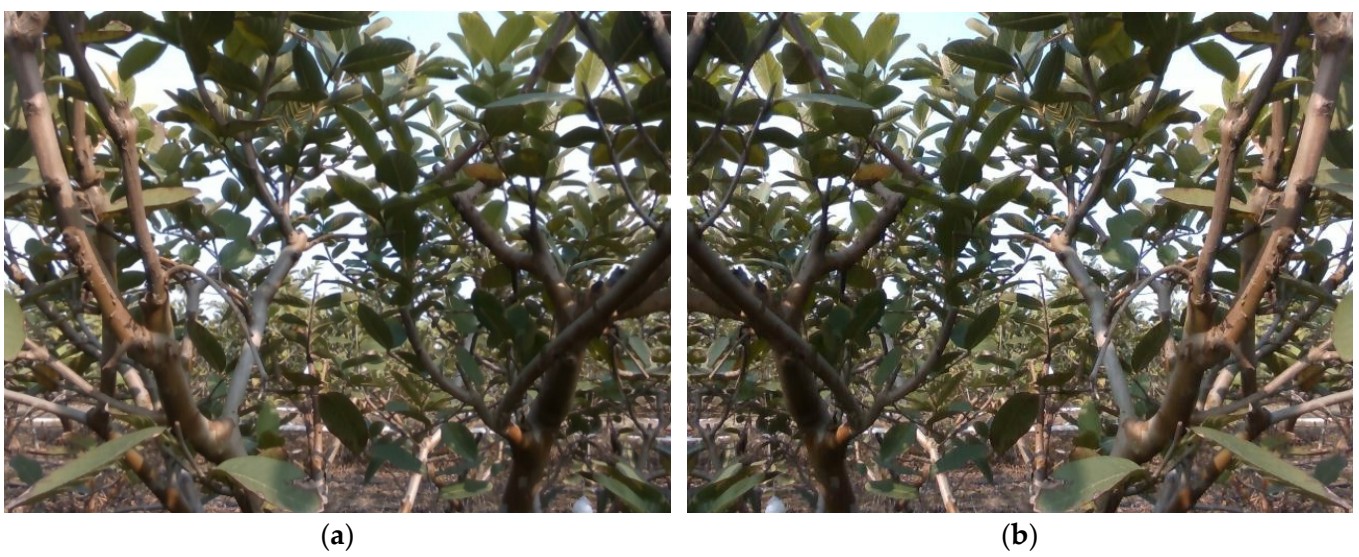

(**a**)　　　　　　　　　　　　　　　　　　　　　　(**b**)

**Figure 3.** Original image (**a**) and its left-right mirroring image (**b**).

### 2.3. Method

Figure 4 shows the network architecture of the developed segmentation model. The model's backbone, Mobilenetv2, uses depth-wise separable convolutional layers to extract different levels of features in real time. In order to improve the semantic and detailed information of the feature maps, FEM is proposed to fuse the information of multi-level features. Then, a decoder module based on self-attention is developed to output a feature map with good segmentation quality.

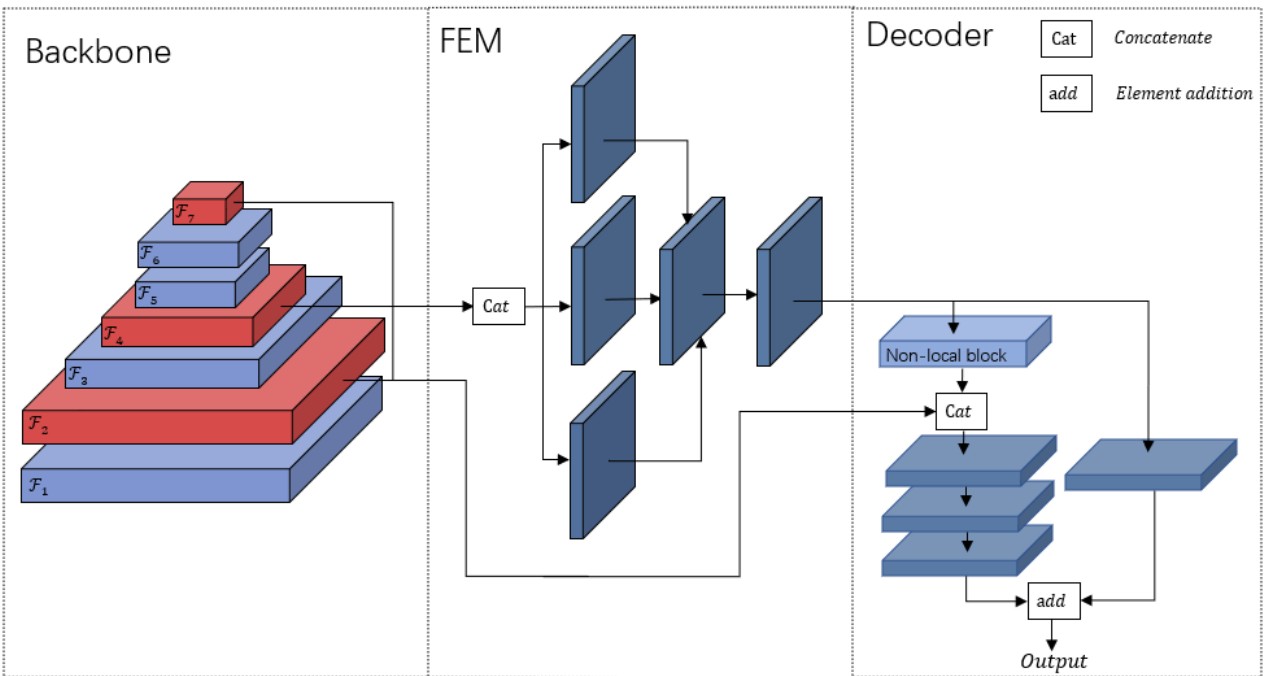

**Figure 4.** Network architecture of the developed model.

### 2.3.1. Backbone

Mobilenetv2 is used as the backbone to generate different levels of feature maps. The network structure of Mobilenetv2 is shown in Table 1, where *c* represents the number of channels; *n* represents the total number of repeated operations; and *s* represents the stride.

In Mobilenetv2, the bottleneck residual block is the basic component, which consists of a $1 \times 1$ ConvBnRelu (CBR) layer, a $3 \times 3$ depth-wise separable convolutional layer, a ReLu activation function and a $1 \times 1$ convolutional layer. Here, CBR consists of three operations: a convolution layer, a batch normalization layer, and a Relu activation function. In this study, $\mathcal{F}_i(i \in (1 \sim 11))$ is defined as the output of the last bottleneck of each line in Table 1.

**Table 1.** Network architecture of Mobilenetv2.

| Input | Operator | c | n | s |
|---|---|---|---|---|
| $224^2 \times 3$ | Conv2d | 32 | 1 | 2 |
| $112^3 \times 32$ | bottleneck | 16 | 1 | 1 |
| $112^2 \times 16$ | bottleneck | 24 | 2 | 2 |
| $56^2 \times 24$ | bottleneck | 32 | 3 | 2 |
| $28^2 \times 32$ | bottleneck | 64 | 4 | 2 |
| $14^2 \times 64$ | bottleneck | 96 | 3 | 1 |
| $14^2 \times 96$ | bottleneck | 160 | 3 | 2 |
| $7^2 \times 160$ | bottleneck | 320 | 1 | 1 |
| $7^2 \times 320$ | bottleneck | 1280 | 1 | 1 |
| $7^2 \times 1280$ | Avgpool $7 \times 7$ | – | 1 | – |
| $1 \times 1 \times 1280$ | bottleneck | k | – | – |

In order to increase the accuracy of obstacle segmentation, the feature maps $\mathcal{F}_2$, $\mathcal{F}_4$ and $\mathcal{F}_7$ are used. The main reasons are: (1) $\mathcal{F}_2$ has a high resolution and thus is filled with detailed information; (2) $\mathcal{F}_7$ has a low resolution but with sufficient information about the global context; (3) $\mathcal{F}_4$ contains a certain number of semantics and details. It should be noted that there are so many channels in the feature maps after $\mathcal{F}_7$, which would significantly increase the FLOPs of the semantic segmentation network and decrease the inference efficiency, so we do not use them.

### 2.3.2. Feature Enhancement Module

As shown in Table 1, the backbone frequently decreases the feature map's resolution, in order to reduce the cost of computing. The resulting low-level feature maps contain detailed information, while the high-level feature maps have strong semantics. Only using the high-level feature maps may has some disadvantages; for instance, the vimineous branches are easily lost in the high-level feature maps, decreasing the segmentation accuracy. To this effect, FEM is proposed to fuse different levels of features to generate a feature map with sufficient details and semantics. Figure 5 shows the architecture of FEM.

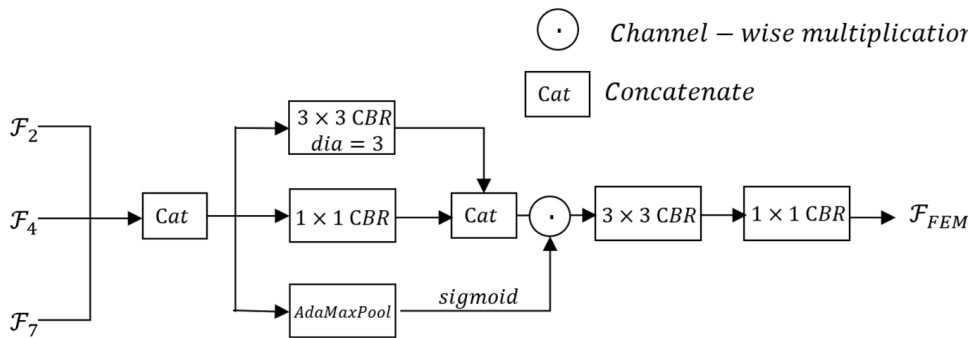

**Figure 5.** Architecture of the developed FEM.

Specifically, the input of FEM is the high-level feature $\mathcal{F}_7 \in \mathbb{R}^{C_7 \times H_7 \times W_7}$, the middle-level feature $\mathcal{F}_4 \in \mathbb{R}^{C_4 \times H_4 \times W_4}$ and the bottom-level feature $\mathcal{F}_2 \in \mathbb{R}^{C_2 \times H_2 \times W_2}$. Because $\mathcal{F}_2$, $\mathcal{F}_4$ and $\mathcal{F}_7$ have different resolutions, $\mathcal{F}_4$ and $\mathcal{F}_7$ are scaled to the same feature map size

as $\mathcal{F}_2$ by bilinear interpolation. Then, $\mathcal{F}_2$, $\mathcal{F}_4$ and $\mathcal{F}_7$ are combined to produce a feature $\mathcal{F}_{concat} \in \mathbb{R}^{(C_2+C_4+C_7) \times H_2 \times W_2}$. This process can be represented by the following formula

$$\mathcal{F}_{concat} = cat(\mathcal{F}_2, \mathcal{F}_4, \mathcal{F}_7) \tag{1}$$

where *cat* refers to the concatenate operation along the channel direction. The quantity of information included in each layer is unaffected by the cat operation here. Therefore, the resulting feature $\mathcal{F}_{concat}$ will contain different detailed and semantic information. Then, this study performs the following two operations on $\mathcal{F}_{concat}$

$$F_{branch} = cat(w_1 * \mathcal{F}_{concat}, w_2 * \mathcal{F}_{concat}) \tag{2}$$

$$\mathcal{F}_{FEM} = sigmoid(AdaMaxPool(\mathcal{F}_{concat}))F_{branch} \tag{3}$$

where $*$ refers to the convolution operation; $w_1$ is a CBR layer with $1 \times 1$ kernel; $w_2$ is a CBR layer with $3 \times 3$ dilated kernel and the dilation rate is 3; and AdaMaxPool refers to the adaptive maximum pooling operation. Specifically, $w_1$ is used to strengthen the connection of each channel, so that the branch information can flow into other channels. The function of $w_2$ is to enlarge the receptive fields of $\mathcal{F}_{concat}$ to obtain more global information. The features of the aforementioned two branches are simply fused together by the *cat* operation to complete the information complementation. The resulting output is denoted as $F_{branch} \in \mathbb{R}^{(C_2+C_4+C_7) \times H_2 \times W_2}$. In order to recalibrate $F_{branch}$ to focus on informative channels, AdaMaxPool is performed on $\mathcal{F}_{concat}$ to obtain $F_{AdaMaxPool} \in \mathbb{R}^{(C_2+C_4+C_7) \times 1 \times 1}$. A sigmoid activation function is then applied to scale $F_{AdaMaxPool}$ to the range of [0, 1]. Subsequently, $F_{AdaMaxPool}$ and $F_{branch}$ are multiplied along their channel dimension. The resulting feature is defined as $\mathcal{F}_{FEM} \in \mathbb{R}^{(C_2+C_4+C_7) \times H_2 \times W_2}$.

Finally, a $3 \times 3$ CBR layer and a $1 \times 1$ CBR layer are performed on $\mathcal{F}_{FEM}$. The $3 \times 3$ CBR layer aims to reduce the number of channels from $C_2 + C_4 + C_7$ to $C(C = 256)$, and increase the receptive field. The $1 \times 1$ CBR layer endeavors to enhance channel connectivity, which can further encourage the flow of information between channels.

FEM brings a total of $5(C_2 + C_4 + C_7)^2 + 9C(C_2 + C_4 + C_7) + C^2$ new parameters, which equals 0.76 million—that is, FEM only adds a small amount of parameters to the backbone. However, FEM increases the computational burden on the backbone to some extent, as most of the computation takes place on the high-resolution feature maps, probably lowering the inference speed.

### 2.3.3. Decoder Module

The decoder module is used to generate an output which is the same size of the input image. This objective could be accomplished by up-sampling $\mathcal{F}_{FEM}$ using deconvolution [22] or bilinear interpolation, however, this might generate coarse results. In order to address this problem, we utilize $\mathcal{F}_{FEM}$ and $\mathcal{F}_2$ to design a novel decoder, as shown in Figure 6. The decoder uses a self-attention layer [23] to strengthen the connection between each position on the feature map $\mathcal{F}_{FEM}$, the output of which is merged with a low-level feature $\mathcal{F}_2$ to improve details. Afterwards, a shortcut connection and element-wise addition is performed to promote the gradient to flow in the network. The details of the decoder are described as follows.

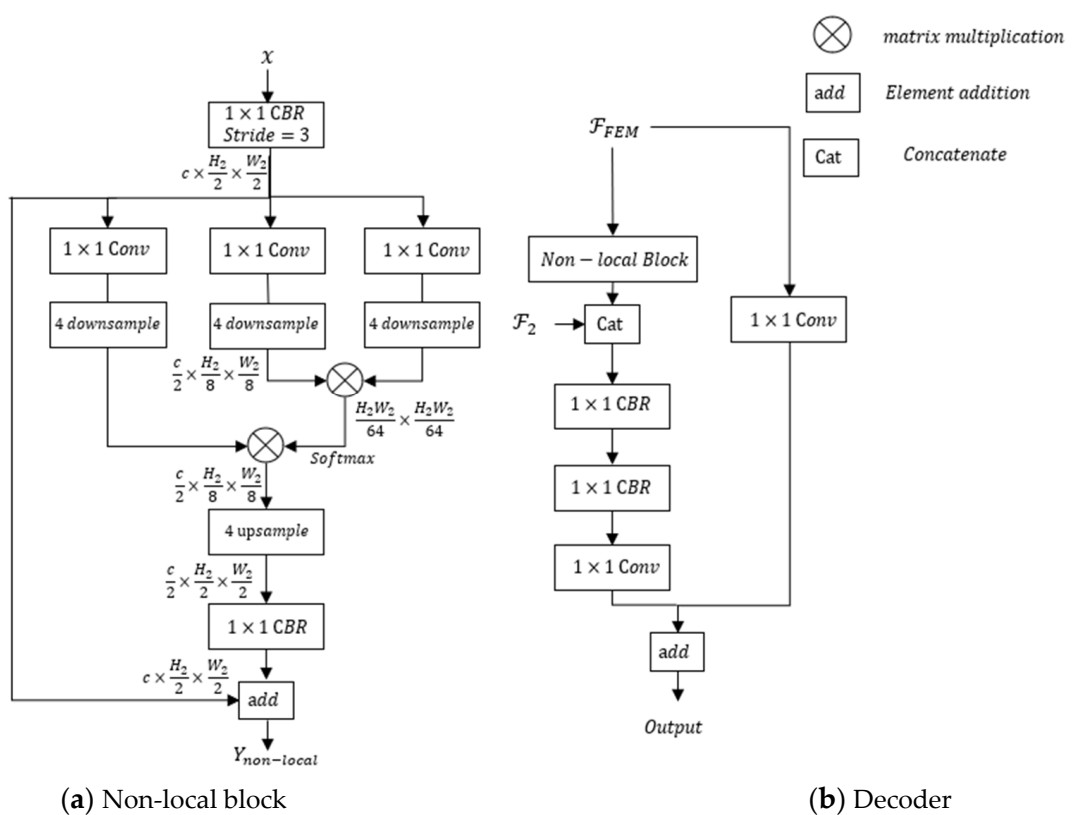

(**a**) Non-local block          (**b**) Decoder

**Figure 6.** Network architecture of the decoder module.

Firstly, the non-local block [23] is used as a self-attention layer, as it can effectively model the non-local relationship between each pixel to capture long-range dependency. In order to reduce the computational cost, the non-local block is modified as shown in Figure 6a: (1) the resolution of $\mathcal{F}_{FEM}$ is first reduced by a $1 \times 1$ CBR with a stride of 2; (2) the number of output channels of the three convolution layers in the non-local block is simply halved; and (3) the outputs of the three convolution layers are down-sampled by bilinear interpolation to further reduce the computation. The resulting feature map is defined as $Y_{non-local} \in \mathbb{R}^{C \times \frac{H_2}{2} \times \frac{W_2}{2}}$.

Secondly, because the vimineous branches are easy to be ignored, $Y_{non-local}$ and $\mathcal{F}_2$ are fused together to increase the detail information. The fusing process is defined as follows:

$$Y_{cat} = cat(\mathcal{F}_2, Y_{non-local}) \tag{4}$$

Note that a bilinear interpolation-based up-sampling is performed on $Y_{non-local}$ before cat.

Thirdly, $Y_{cat} \in \mathbb{R}^{(C+C_2) \times H_2 \times W_2}$ passes through two CBR layers, each with a kernel size of $1 \times 1$. The first layer is mainly to reduce the dimension by changing the number of channels from $C + C_2$ to $C$. The second layer concentrates on improving the connections between different channels and promoting cross-channel information interaction. The resulting feature map is defined as $\mathcal{F}_{FEM2}$. Both $\mathcal{F}_{FEM2}$ and $\mathcal{F}_{FEM}$ are further processed by a $1 \times 1$ convolution layer and merged together by element-wise addition, which can be regarded as a residual block that can promote the gradient to flow in the network. The merged feature map is up-sampled by bilinear interpolation to output the final prediction.

### 2.3.4. Loss Function

In the guava dataset, the background dominates the images—that is, the data is extremely unbalanced. To avoid the network to focus on the background, it is vital to

choose an appropriate loss function. In this study, we combine Focal loss [24] and Dice loss [25] as the final loss function:

$$loss = Loss_{FL} + Loss_{De} \tag{5}$$

Focal loss has an excellent performance for dealing with extremely foreground-background class imbalance, which is defined as follows

$$Loss_{FL} = -\alpha\gamma(1-p)^{\gamma} \cdot log(p) - (1-\alpha)(1-y)p^{\gamma} \cdot log(1-p) \tag{6}$$

when $\alpha$ is a constant and $\gamma$ represents the focusing parameter. Dice loss calculates the intersection ratio between the predicted value and the ground truth, which is defined as follows

$$Loss_{De} = 1 - \frac{2|X \cap Y|}{|X| + |Y|} \tag{7}$$

where $X$ represents the ground truth and $Y$ represents the prediction.

### 2.3.5. Implementation Details

We utilize two Titan RTX cards with 24 GB RAM, an Intel Gold 5218 processor and an Ubuntu system to train and test the network. The version of CUDA is 11.2. The network is implemented in PyCharm with PyTorch. In the training phase, a pre-trained Mobilenetv2 on ImageNet is used as the backbone. The backbone's initial learning rate is set at $1 \times 10^{-4}$, while that of the other modules is $1 \times 10^{-3}$. A cosine annealing strategy with a period of 200 is adopted to adjust the learning rates. The batch size is set to 16. An Adam optimizer with weight decay of $5 \times 10^{-4}$ is used.

### 2.4. Evaluating Indicators

In this paper, MIOU, MPA and FWIOU are used as the precision evaluation indexes. FLOPs, FPS and the number of parameters are used as the speed evaluation indexes. MIOU, MPA and FWIOU are defined as follows

$$\text{MIOU} = \frac{1}{N+1} \sum_{i=1}^{N} \frac{p_{ij}}{\sum_{j=0}^{N} p_{ij} + \sum_{j=0}^{N} p_{ij} - p_{ii}} \tag{8}$$

$$\text{MPA} = \frac{1}{N+1} \sum_{i=1}^{N} \frac{p_{ij}}{\sum_{j=0}^{N} p_{ij}} \tag{9}$$

$$\text{FWIOU} = \frac{1}{\sum_{i=0}^{N} \sum_{j=0}^{N} p_{ij}} \sum_{i=0}^{N} \frac{\sum_{j=0}^{N} p_{ij} p_{ii}}{\sum_{j=0}^{N} p_{ij} + \sum_{j=0}^{N} p_{ij} - p_{ii}} \tag{10}$$

where $N$ represents the number of classification classes; and $p_{ij}$ is the probability that $i$ will turn out to be $j$. $p_{ii}$ represents the probability that $i$ is predicted to be accurate.

## 3. Results

### 3.1. Ablation for FEM

The purpose of this experiment was to explore the effect of fusion between different layers, and to verify the performance under various combinations. Since $\mathcal{F}_2$ and $\mathcal{F}_7$ contains sufficient details and semantic information, respectively, we designed six experiments based on $\mathcal{F}_2 + \mathcal{F}_7$. The experimental results on the test set were shown in Table 2.

**Table 2.** Accuracy and real-time performance of different FEMs. **M** is million, and **B** is billion.

| Combination | MIOU | MPA | FWIOU | Params (M) | FLOPs (B) | FPS |
|---|---|---|---|---|---|---|
| $\mathcal{F}_2 + \mathcal{F}_7$ | 75.55 | 83.71 | 88.87 | 3.6 | 29.3 | 47.4 |
| $\mathcal{F}_2 + \mathcal{F}_7 + \mathcal{F}_3$ | 75.68 | 85.03 | 88.92 | 3.7 | 32.4 | 46.6 |
| $\mathcal{F}_2 + \mathcal{F}_7 + \mathcal{F}_4$ | 76.30 | 84.63 | 89.04 | 3.9 | 35.7 | 45.9 |
| $\mathcal{F}_2 + \mathcal{F}_7 + \mathcal{F}_5$ | 75.63 | 85.09 | 88.77 | 4.2 | 39.1 | 45.5 |
| $\mathcal{F}_2 + \mathcal{F}_7 + \mathcal{F}_6$ | 75.22 | 83.97 | 88.88 | 4.6 | 46.5 | 44.1 |
| $\mathcal{F}_2 + \mathcal{F}_3 + \mathcal{F}_4 + \mathcal{F}_5 + \mathcal{F}_6 + \mathcal{F}_7$ | 75.78 | 84.59 | 88.96 | 6.2 | 72.0 | 37.6 |

The table shows that $\mathcal{F}_2 + \mathcal{F}_7$ was the fastest but a bit less accurate; $\mathcal{F}_2 + \mathcal{F}_7 + \mathcal{F}_4$ performed the best in MIOU and FWIOU, while being quite fast in terms of Params, FLOPS and FPS; $\mathcal{F}_2 + \mathcal{F}_7 + \mathcal{F}_5$ only outperformed the others in MPA, and had a lower speed than $\mathcal{F}_2 + \mathcal{F}_7 + \mathcal{F}_4$. Furthermore, the results of $\mathcal{F}_2 + \mathcal{F}_3 + \mathcal{F}_4 + \mathcal{F}_5 + \mathcal{F}_6 + \mathcal{F}_7$ revealed that with the fusion of more feature maps, the accuracy of the network was declining, and the inference time was increasing. As $\mathcal{F}_2 + \mathcal{F}_7 + \mathcal{F}_4$ has a high accuracy and reasonable inference time, $\mathcal{F}_2 + \mathcal{F}_7 + \mathcal{F}_4$ was used by FEM in the following experiments.

*3.2. Ablation for Loss Function*

The convergence and effectiveness of the model were significantly impacted by the loss function selection. CE loss, Focal loss, Dice loss, and Focal loss + Dice loss were used as the loss functions for experiments, where CE loss was defined as

$$Loss_{CE} = -\sum_{i=1}^{N} p_i \log(q_i) \tag{11}$$

where $N$ represents the number of classes; $p_i$ is the probability value of the real output; and $q_i$ is the probability value of the predicted output.

Table 3 shows the experimental results on the test set. CE loss performed best in terms of MPA and FWIOU, while Focal loss + Dice loss performed best in terms of MIOU, and has a similar performance as CE loss in MPA. The performance of Focal loss and Dice loss was the worst.

**Table 3.** Effects of different loss functions.

| Loss Function | MIOU | MPA | FWIOU |
|---|---|---|---|
| CE Loss | 76.22 | 84.65 | 89.68 |
| Focal Loss | 74.46 | 83.90 | 89.17 |
| Dice Loss | 75.53 | 84.25 | 89.19 |
| Focal Loss + Dice Loss | 76.30 | 84.63 | 89.04 |

*3.3. Comparison with Other Methods*

PSPnet and Deeplabv3+ were utilized as the baselines for comparisons. Fairness was ensured by using Mobilenetv2 as the backbone for PSPnet and Deeplabv3+, and employing the training approach described in Section 2.3.5. Table 4 displays the experimental results on the test set. Some visual examples are depicted in Figure 7.

**Table 4.** Comparison with state-of-the-art methods.

| Method | MIOU | MPA | FWIOU | Param (M) | Flops (B) | FPS |
|---|---|---|---|---|---|---|
| PSPnet | 72.53 | 82.20 | 87.34 | 2.3 | 2.9 | 70.6 |
| Deeplabv3+ | 74.47 | 83.03 | 88.61 | 5.8 | 26.5 | 53.4 |
| Our model | 76.30 | 84.63 | 89.04 | 3.9 | 35.7 | 45.0 |

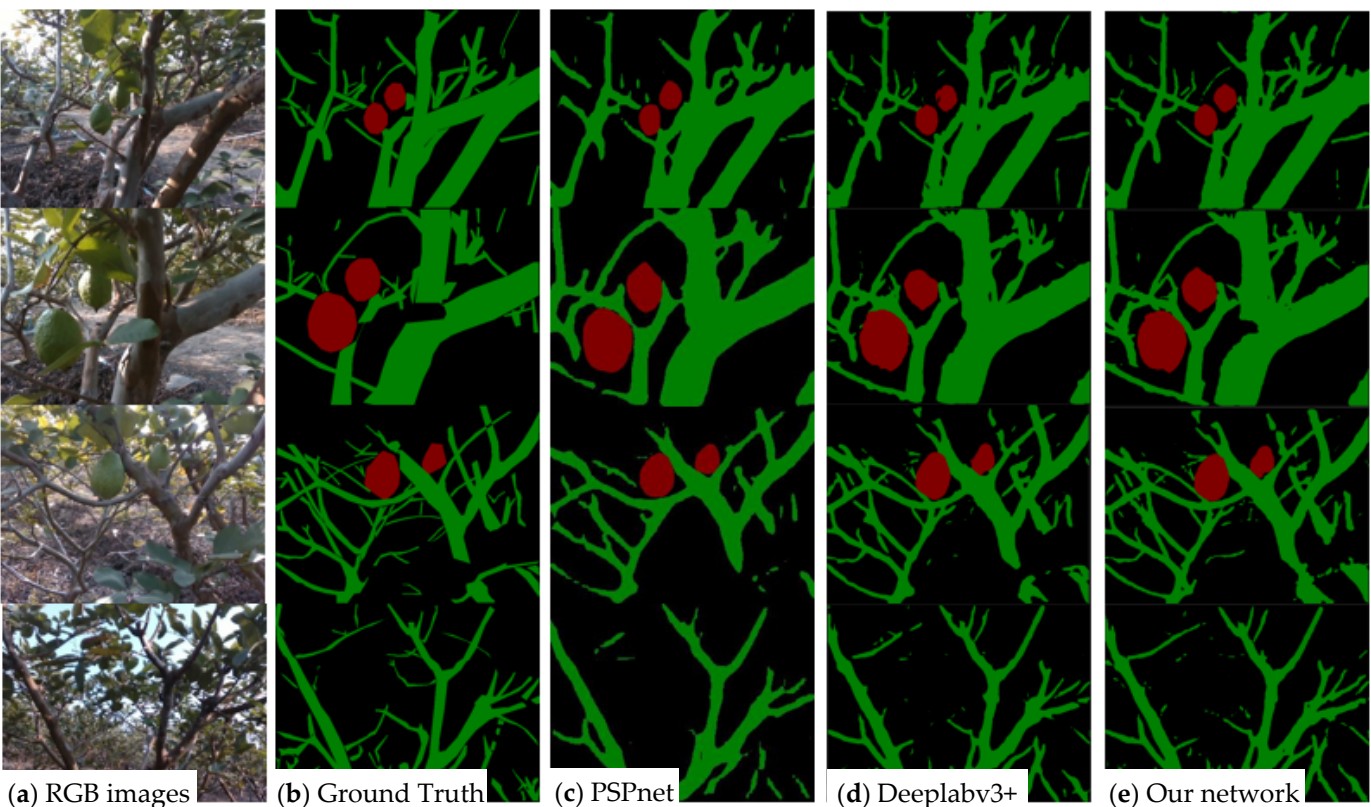

(**a**) RGB images    (**b**) Ground Truth    (**c**) PSPnet    (**d**) Deeplabv3+    (**e**) Our network

**Figure 7.** Segmentation results of different networks.

As can be seen from Table 4, our network was more accurate than Deeplabv3+ and PSPnet in terms of MIOU, MPA and FWIOU. However, our network was a bit slower than Deeplabv3+ and PSPnet in terms of FLOPs and FPS. In summary, our network strode a trade-off between accuracy and speed.

Figure 7 shows that all the networks could segment the fruits accurately, and our network outperformed the comparison networks in segmenting the branches, especially the vimineous branches.

## 4. Discussion

### 4.1. Ablation for FEM

As shown in Table 2, the model's segmentation accuracy could be increased by incorporating additional intermediate layers, as the intermediate layer contained sufficient semantic and detailed information. It was obvious that different intermediate layers have different effects on network performance. That is, the semantic information of $\mathcal{F}_5$ or $\mathcal{F}_6$ was more than their detailed information, causing the semantic to be over enhanced and the detailed to be under enhanced; the detailed information of $\mathcal{F}_3$ was more than their semantic information, causing the detailed to be over enhanced and the semantic to be under enhanced. Experiment results show that $\mathcal{F}_4$ balanced the semantic and detail information of FEM, and the best combination was $\mathcal{F}_2 + \mathcal{F}_7 + \mathcal{F}_4$.

Additionally, increasing the number of intermediate layers in FEM decreased the accuracy unexpectedly. A possible reason was that with more intermediate layers merged, FEM needed wider convolution kernels to fuse and recalibrate the merged feature map, which made the network hard to train. Therefore, when there were too many intermediate layers in FEM, the network accuracy would decrease. Moreover, since FEM put most of its computation on the high-resolution feature map, using too many intermediate layers would significantly lower the inference speed.

### 4.2. Ablation for Loss Function

The combination of Focal loss + Dice loss worked best in terms of MIOU, and was almost as well as CE loss in MPA, but performed poorly in FWIOU. The performance of Focal loss + Dice loss on FWIOU was normal. FWIOU estimates the weight of each category by calculating the occurrence frequency of the categories. So, a category would have a large weight if it occurs more frequently. Because our dataset contained a large number of backgrounds that dominate the images, the fruit and obstacle categories had a relatively small weight, making the FWIOU a little biased. Future work should consider using a more reasonable evaluation indicator.

### 4.3. Comparison with Other Methods

As seen in Table 4, in terms of the accuracy metrics like MIOU, MPA and FWIOU, our network performed best followed by Deeplabv3+ and PSPnet. The reason why PSPnet performed worst was that PSPnet did not utilize multi-level features, and instead immediately up-sampled the coarsest feature map to output a low-accuracy segmentation map. The performance of Deeplabv3+ ranked second, as Deeplabv3+ increased the detailed information of the coarse feature map by fusing low-level features. Overall, the results validated the effectiveness of the developed FEM and decoder. Unfortunately, Figure 7 shows that some vimineous branches were hard to segment by our network, likely because Mobilenetv2 was too lightweight to provide enough detailed and semantic information.

In terms of the speed metrics like FLOPs and FPS, our network was a bit more time-consuming than Deeplabv3+ and PSPnet, probably because FEM and the decoder put most of their computation on the high-resolution feature maps, which significantly increased the computation. A fast inference speed allows the model to be deployed on a low-cost edge computer that probably only requires a small amount of power, and thus increases the operation time of the robot. Therefore, in order to further improve the inference speed, future work should focus on how to optimize FEM and the decoder.

### 5. Conclusions

The study investigates a fast and accurate obstacle segmentation network for guava-harvesting robots. The results of the experiments demonstrate that the network was capable of precisely and quickly segmenting the branches and fruits in challenging situations. The following findings were drawn from this research:

1. A feature merging and enhancement module was proposed to generate a feature map with strong semantics and details. Experiment results reveal that fusing as many features as possible would decrease the segmentation accuracy and slow down the inference speed; the best combination was $\mathcal{F}_2 + \mathcal{F}_7 + \mathcal{F}_4$.
2. A decoder module was developed by using a self-attention layer to capture a long-range dependency for every pixel in the feature map, and utilizing a shortcut connection and element-wise addition to promote the gradient to flow in the network, thus improving the segmentation accuracy.
3. Our network's MIOU, MPA and FWIOU were 76.30%, 84.63% and 89.04%, respectively, which were 1.83%, 1.60% and 0.43% higher than deeplabv3+, and 3.77%, 2.43% and 1.70% higher than PSPnet. In addition, our network achieved an inference speed of 45 FPS. The results revealed that the model can accurately and quickly segment obstacles for the guava picking robots.

Future work will explore the following problems: (1) how to improve the segmentation accuracy for vimineous branches; and (2) how to reduce the computational burden in FEM and the decoder.

**Author Contributions:** Conceptualization, J.Y. and Q.Y.; methodology, J.Y.; software, G.D., T.W. and D.Z.; validation, J.Y., G.D. and T.W.; formal analysis, Q.Y.; investigation, G.L.; resources, P.H.; data curation, G.D. and T.W.; writing—original draft preparation, J.Y.; writing—review and editing, G.L.

and L.Z.; visualization, G.L.; supervision, L.Z.; project administration, L.Z.; and funding acquisition, G.L., P.H. and L.Z. All authors have read and agreed to the published version of the manuscript.

**Funding:** This work was funded by the Laboratory of Lingnan Modern Agriculture Project (Grant No. NZ2021038), the National Natural Science Foundation of China (Grant No. 32101632), the Basic and Applied Basic Research Project of Guangzhou Basic Research Plan (Grant No. 202201011310; 202201011691) and the Science and Technology Program of Meizhou, China (Grant No. 2021A0304004).

**Institutional Review Board Statement:** Not applicable.

**Informed Consent Statement:** Not applicable.

**Data Availability Statement:** Data recorded in the current study are available in all tables and figures of the manuscript.

**Conflicts of Interest:** The authors declare no conflict of interest.

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
