# Peer review of "A Fast and Accurate Obstacle Segmentation Network for Guava-Harvesting Robot via Exploiting Multi-Level Features"

_sustainability, doi:10.3390/su141912899_

Round 1
Reviewer 1 Report
The article is written clearly, but the abstract is recommended to be revised. The article is devoted to the actual problem of technical vision in agriculture. However, the text contains significant stylistic errors. In particular, a sentence of 60 words makes it difficult to understand the idea being expressed. Such a sentence of 60 words should be broken into 2 or 3 sentences. The abstract of the article talks about fast data processing, but does not indicate the quantitative parameters of image processing speed. The article needs to be improved.
Reviewer 2 Report
Dear Editor/s
The article entitled: A fast and accurate obstacle segmentation network for guava-2 harvesting robot via exploiting multi-level features, has been reviewed carefully. It is a good written and interesting paper which appropriate to publish in Journal of sustainability-1888683. However there are some questions and recommendation to improve it before publication:
1- Page 2- Line: 88-91
what was the distance between trees and camera when you were taking a picture?
what was the size of Image Matrix?
2- Page 3- Line 96:
what was the software name?? and version?
3- Page 10-Line 277
please show a, b, ... on the pictures
4- Page 10- Line 286:
in this section, is better to compare your results with other works which published
5- A General question:
What were the advantages or disadvantages of your work comparing with the other researcher’s results?
Please make a discussion
6- What was the main novelty of your work and how to implement your results in the field?

Reviewer 3 Report
First I want to congratulate you for the work you have done. The paper present an obstacle segmentation network for guava-harvesting robot using a set of advanced algorithms which are demanding high computational resources. I think it will be intersting if the authors will present also the computing resources utilisation during the execution of the algorithms and the power demand.
Also it will be interesting to add a discution on the way that the authors want to integrate the system on the robot (using online services, using a completly autonomous robot), how many hours of operation per day are targeted, power consumption, battery capacity, etc.
